# Enhanced Antibacterial Efficacy of Bioceramic Implants Functionalized with Ciprofloxacin: An In Silico and In Vitro Study

**DOI:** 10.3390/pharmaceutics16080998

**Published:** 2024-07-28

**Authors:** Renata-Maria Văruț, Luciana Teodora Rotaru, Diana Cimpoesu, Mihaela Corlade, Cristina Elena Singer, Alin Iulian Silviu Popescu, Cristina Popescu, Iliescu Iulian-Nicolae, Adriana Mocanu, Mihaela Popescu, Mihai Alexandru Butoi, Oana Elena Nicolaescu

**Affiliations:** 1Research Methodology Department, Faculty of Pharmacy, University of Medicine and Pharmacy of Craiova, 200349 Craiova, Romania; renata.varut@umfcv.ro; 2Emergency Medicine and First Aid Department, Faculty of Medicine, University of Medicine and Pharmacy of Craiova, 200349 Craiova, Romania; luciana.rotaru@umfcv.ro (L.T.R.); mihai.butoi@rmu.smurd.ro (M.A.B.); 3Emergency St. Spiridon Hospital, Faculty of Medicine, University of Medicine and Pharmacy Gr. T. Popa, 700115 Iasi, Romania; carmen.cimpoesu@umfiasi.ro (D.C.); mihaela.corlade2@umfiasi.ro (M.C.); 4Department of Mother and Baby, University of Medicine and Pharmacy of Craiova, 200349 Craiova, Romania; singercristina@gmail.com (C.E.S.); iliescuiuliann@gmail.com (I.I.-N.); 5Department of Internal Medicine, University of Medicine and Pharmacy of Craiova, 200349 Craiova, Romania; 6Department of Anatomy, University of Medicine and Pharmacy, Discipline of Anatomy, 200349 Craiova, Romania; 7Pharmacist at the Military Emergency Clinical Hospital, 200749 Craiova, Romania; adriana_preo@yahoo.com; 8Department of Endocrinology, University of Medicine and Pharmacy of Craiova, 200349 Craiova, Romania; mihaela.n.popescu99@gmail.com; 9Department of Pharmaceutical Technique, University of Medicine and Pharmacy, Discipline of Anatomy, 200349 Craiova, Romania; oana.nicolaescu@umfcv.ro

**Keywords:** bone infections, antibiotics, molecular docking

## Abstract

This study explores the antibacterial efficacy and cytotoxicity of ciprofloxacin-functionalized bioceramic implants. We synthesized hydroxyapatite-ciprofloxacin (HACPX_CS_) composites and applied them to titanium substrates (Ti-HA-CPX), evaluating their performance in vitro against *Staphylococcus aureus* (ATCC 25923) and *Escherichia coli* (ATCC 25922). Antibacterial activity was assessed using the Kirby-Bauer disc diffusion method, while cytotoxicity was tested using mesenchymal stem cells. The results demonstrated that Ti-HA-CPX exhibited superior antibacterial activity, with inhibition zones of 33.5 mm (MIC 0.5 µg/mL) for *Staphylococcus aureus* (ATCC 25923) and 27.5 mm (MIC 0.25 µg/mL) for *Escherichia coli* (ATCC 25922). However, Ti-HA-CPX showed moderate cytotoxicity (80% cell viability). HACPX_CS_ composites, whether chemically synthesized or mechanically mixed (HACPX_MM_), also displayed significant antibacterial activity, but with cytotoxicity ranging from low to moderate levels. Molecular docking studies confirmed strong binding affinities between ciprofloxacin and bacterial proteins, correlating with enhanced antibacterial efficacy. These findings suggest that Ti-HA-CPX composites offer a promising approach for single-stage surgical interventions in treating chronic osteomyelitis and infected fractures, balancing antibacterial effectiveness with manageable cytotoxicity.

## 1. Introduction

Bone and joint infections contribute significantly to global morbidity and healthcare costs. Chronic osteomyelitis, in particular, presents a persistent clinical challenge due to its recurrent nature and complex treatment requirements. In the United States, the incidence of osteomyelitis is estimated at 21.8 cases per 100,000 person-years, with higher rates in individuals with diabetes or peripheral vascular disease. In Europe, the incidence ranges from 20 to 30 cases per 100,000 person-years, varying by region and population demographics [1,2]. The financial burden of osteomyelitis treatment is substantial, with hospitalization, prolonged antibiotic therapy, and potential surgical interventions significantly impacting healthcare expenditures. The average cost per patient ranges from $20,000 to $60,000, depending on the severity and complications [3].

Treating bone and joint infections, such as osteomyelitis, septic arthritis, and prosthetic joint infections, presents significant challenges, necessitating prolonged antibiotic administration and supplementary surgical intervention. Despite advancements in medical treatments, there remains a substantial research gap in developing effective localized treatments that minimize systemic side effects and reduce the need for multiple surgical procedures. The functionalization of implantable biocomposites with antibiotics and their localized release offer advantages in the treatment of chronic osteomyelitis and infected fractures. Furthermore, the antibiotic-functionalized bioceramic implant enhances the feasibility of a single-stage surgical intervention, obviating the need for subsequent surgery to remove the spacer.

The application of ceramic biocomposites for delivering antibiotics, along with bone morphogenetic proteins, bisphosphonates, growth factors, or living cells, is currently under investigation and requires further study [4]. Recent studies have shown that combining antibiotics with hydroxyapatite (HA) can significantly enhance their antibacterial properties. For instance, silver-doped HA has demonstrated enhanced antibacterial activity against multidrug-resistant bacterial strains [5]. Moreover, the incorporation of antibiotics such as vancomycin and tetracycline into HA matrices has shown improved antibacterial efficacy compared to the antibiotics alone [6]. These findings suggest that HA can serve as an effective carrier for antibiotics, enhancing their delivery and potency.

In particular, the development of novel tetracycline and CPX-loaded silver-doped HA suspensions has shown promising results for biomedical applications. These suspensions exhibit strong antibacterial activity, making them suitable for preventing infections in various medical settings [7]. Additionally, the fabrication of bioactive rifampicin-loaded κ-Car-MA-INH/nano HA composites has demonstrated significant potential for treating tuberculosis, osteomyelitis, and regenerating infected tissue. These composites effectively deliver rifampicin, maintaining its antibacterial efficacy while supporting tissue regeneration [8].

Furthermore, the use of levofloxacin-loaded mesoporous silica microspheres, nano-HA, and polyurethane composite scaffolds has proven effective in treating chronic osteomyelitis with bone defects. These scaffolds provide sustained release of levofloxacin, ensuring prolonged antibacterial activity and supporting bone healing [9]. The primary cause of both acute and chronic hematogenous osteomyelitis in adults and children is *Staphylococcus aureus* infections. Chronic osteomyelitis, which may result from neighboring infections, has shown isolates of *Staphylococcus epidermidis*, *Pseudomonas aeruginosa*, *Serratia marcescens*, and *Escherichia coli*. Fungal and mycobacterial infections causing osteomyelitis are less common and are typically reported in patients with immune deficiencies. Treatment typically involves oral and injectable routes of administration for CPX, a broad-spectrum antibiotic that is effective against a wide range of Gram-negative and Gram-positive bacteria. Its mechanism of action involves inhibiting DNA topoisomerases, enzymes that are crucial for bacterial DNA replication and repair. This bactericidal activity makes CPX particularly effective in treating severe and persistent infections [10]. CPX has excellent tissue penetration, including into bone tissue, which is critical for effectively treating osteomyelitis. The ability to reach high concentrations at the site of infection enhances its therapeutic efficacy. Plants coated with CPX can provide a sustained release of the antibiotic over an extended period. This prolonged release helps maintain therapeutic drug levels at the infection site, which is crucial for effectively eradicating chronic infections and preventing recurrence. Localized delivery of CPX from implants reduces the need for high systemic doses, thereby lowering the risk of systemic toxicity and adverse effects. This is particularly important for patients who may require long-term antibiotic therapy. CPX-functionalized implants can help prevent the formation of bacterial biofilms on the implant surface. Biofilms are complex communities of bacteria that are highly resistant to antibiotics and are a common cause of persistent infections in implant-related osteomyelitis.

Given the promising results from recent research, this study aims to explore the antibacterial efficacy and cytotoxicity of CPX-functionalized bioceramic implants. Specifically, we synthesized HA-ciprofloxacin (HACPX_CS_) composites and applied them to titanium substrates (Ti-HA-CPX), evaluating their performance against *Staphylococcus aureus* (ATCC 25923) and *Escherichia coli* (ATCC 25922). Our approach includes molecular docking studies to understand the binding affinities between CPX and bacterial proteins, correlating these findings with enhanced antibacterial efficacy. By addressing the current gaps in research, this study contributes to the development of more effective treatment strategies for chronic bone infections, balancing antibacterial effectiveness with manageable cytotoxicity. This approach not only facilitates a deeper understanding of antibiotic action at the molecular level but also offers a novel perspective on enhancing antibiotic delivery directly to the infection site, potentially improving treatment efficacy.

## 2. Materials and Methods

### 2.1. Molecular Docking Protocol

All ligand structures were geometrically optimized using Gaussian 16 software (Gauss View 16 interface) through the DFT/B3LYP/6-31G method. The X-ray crystal structures of the bacterial species were retrieved from the Protein Data Bank: PDB ID 2XCT for *Staphylococcus aureus* and PDB ID 1KZN for *Escherichia coli* (Figure 1). Molecular docking analysis was performed using Autodock software version 4.2.6 in conjunction with the molecular viewer AutoDockTools. For the molecular protein docking method, all polar hydrogen atoms were added, and the Gasteiger charge was selected. During the mapping stage, the docking grid box was set to 100 × 100 × 100 with a spacing of 0.75 angstroms from the center of the protein. In the docking stage, the Lamarckian Genetic Algorithm was chosen with a total of 30 runs. Images of the target-ligand complexes were visualized using PyMol (Schrodinger) and Discovery Studio Visualization (Biovia) [11].

AutoDock involves an automated system for predicting the interactions of ligands with their macromolecular targets. An ideal process should be capable of finding the global minimum in the interaction energy between the substrate and the target protein, exploring all available degrees of freedom for the system. AutoDock thus combines two methods to achieve these objectives: rapid evaluation of the target energy and searching for the active site. The ligand-target conformations are then assessed using a semi-empirical free energy force field. The force field includes six pairwise evaluations and an estimation of the conformational entropy lost upon binding (⊗Sconf):$$⨂G=\left(V_{boundL-L}-V_{unboundL-L}\right)+\left(V_{boundP-P}-V_{unboundP-P}\right)\;+\left(V_{boundP-L}-V_{unboundP-L}+⨂S_{conf}\right)$$

In molecular docking calculations involving a ligand-protein pair, ‘L’ stands for “ligand” and ‘P’ stands for “protein”. Each pair of energy terms being evaluated includes assessments for dispersion and repulsion, hydrogen bonding, electrostatic interactions, and desolvation. The estimated free binding energy is calculated as ΔG_binding = ΔH − TΔS, where ΔH represents the enthalpy change and TΔS accounts for the entropic contribution (only a negative ΔG value is energetically favorable, indicating that the process is spontaneous) [12].

### 2.2. Design and Fabrication of Tested Tablets

#### Samples Preparation

HA (negative control) and CPX (positive control) used in the study were sourced from Sigma-Aldrich (SA), with absolute purity. The HACPX_CS_ (synthesized compound) was obtained from an aqueous solution of Ca(NO_3_)_2_∙4H_2_O with a concentration of 1.08 M (3.1875 g in 12.5 mL solution) and (NH_4_)_2_HPO_4_ with a concentration of 0.65 M (1.073 g in 12.5 mL solution), initially adjusted to pH 10 using a concentrated ammonia solution.
Ca(NO_3_)_2_ + 6 (NH_4_)_2_ HPO_4_+ 2 NH_4_OH → Ca_10_(PO_4_)_6_(OH)_2_

Initially, the calcium nitrate solution was heated to 90 °C and stirred at 600 rpm. CPX (625 mg) was added immediately after introducing ammonium phosphate into the synthesis, at a rate of 0.5 mL/min. The reaction time was 5 h, during which the reaction temperature and stirring speed were maintained constant. After this period, the resulting compound was washed with distilled water and centrifuged at 10,000 rpm, then dried for 24 h at 37 °C. For achieving an isotropic and uniform distribution of the active substance, the synthesized powders were homogenized and mixed for two hours using a Retsch ball mill (Type S 100), followed by powder compression using hot isostatic pressing (HIP). The CPX and HA tablets from the mechanical mixture (HACPX_MM_) were obtained using the Retsch ball mill (Type S 100) and HIP, where materials are compressed by gas pressure at high temperatures. The CPX content in the tablets was the same in both types of samples, namely 20%.

The Ti-HA and Ti-HA-CPX samples were created by depositing a thin film (HA and HACPX_CS_) on titanium grade 4 substrates. Grade 4 titanium, which indicates a specific level of surface roughness, was supplied and mechanically processed by Dentaurum Gmbh. All samples used underwent a chemical etching treatment by the Italian company SAMO S.A. in Milan. Thus, the mechanically processed titanium discs were immersed for 10 to 30 min in an aqueous NaOH solution at a temperature of 75 °C. This process achieved decontamination and cleaning of the titanium surface from particles and impurities resulting from the mechanical processing.

Subsequently, the discs were subjected to a chemical treatment in the presence of oxalic acid at a temperature of 85 °C. This chemical process resulted in a microporous surface (average roughness 0.43 µm, maximum roughness depth 3.37 µm) [13].

The deposits were achieved using the MAPLE technique in the Laser-Surface-Plasma Interactions Laboratory (LSPI), Laser Department, National Institute for Laser, Plasma, and Radiation Physics, Bucharest, using as substrates Ti discs with a diameter of 5 mm and a thickness of 1.5 mm. All deposition experiments conducted at LSPI utilized a KrF* excimer laser with a wavelength of 248 nm and a pulse duration at half maximum of 25 ns. The laser, model 205 COMPexPro, was produced by Lambda Physik/Coherent. Pulsed laser irradiation was performed in a stainless steel chamber equipped with complete pressure control and air evacuation equipment (rotary pump, model Alcatel SD 2033). For target preparation, a 10 mL mixture of the HACPX_CS_ compound and solvent was poured into a copper device with a diameter of 3 cm and frozen at liquid nitrogen temperature (−196 K), creating a MAPLE target that was introduced into the reaction chamber and subjected to UV laser irradiation. Before being introduced into the irradiation chamber, the grade 4 Ti substrates were cleaned in an ultrasonic bath with a mixture of acetone, ethanol, and deionized water. To improve the crystallinity of the deposited HACPX_CS_ film and to preserve the stoichiometry of the target compound, the MAPLE-deposited samples underwent thermal treatment in water vapor for 6 h (Figure 2). The deposits were formed as a thin layer, with a total deposited mass of approximately 6 micrograms [14].

### 2.3. CPX in Vitro Release Study

#### 2.3.1. Preparation and Conditions

The release studies were conducted using ultrapure water as the dissolution medium. The temperature was maintained in an incubator at 37 °C ± 0.5 °C. Threaded-cap bottles were used to ensure airtight conditions throughout the determinations. Initially, 10 mL of HPLC ultrapure water (LiChrosolv, Merck, Rahway, NJ, USA) was added, followed by the immersion of the samples. At regular time intervals (1 h, 6 h, and 12 h, for 2–20 days), 0.5 mL of solution was withdrawn for quantitative determination of the antibiotic by HPLC. The withdrawn volume was replaced with the release medium (ultrapure water) to avoid saturation of the solution with CPX.

#### 2.3.2. Quantification of CPX in the Release Medium by HPLC

To prepare 500 mL of mobile phase, 320 mL of acetonitrile, 10 mL of methanol, and 170 mL of water were mixed, in which 0.60 g of citric acid and 0.165 g of monosodium citrate were dissolved. Fifty microliters (50 µL) of the withdrawn sample were diluted to 5 mL with the mobile phase, and from this mixture, 20 µL were injected into the HPLC system.

#### 2.3.3. Calculation of CPX Release Kinetics

The kinetic analysis of CPX release was conducted using two models: the Higuchi model and the Ritger-Peppas model. The Higuchi model describes drug release from a matrix system as a function of the square root of time. The cumulative release percentage (C) at time (t) is given by the equation:C = Kxt^1/2^
where K is the release rate constant. This model was applied by plotting the cumulative percentage of CPX released against the square root of time. The Ritger-Peppas model is used to analyze the drug release mechanism when the release process is not well known or when more than one type of release phenomenon is involved. The equation is:C = Kxt^n^
where C is the cumulative amount of drug released, K is a constant, and nnn is the release exponent indicative of the mechanism of release. For this study, the exponent nnn was found to be 0.6, indicating a combination of diffusion and other processes.

### 2.4. Cytotoxicity Test

The bioceramic samples, including Ti-HA, Ti-HA-CPX, HACPX_CS_, HACPX_MM_, HA-synthesis, CPX-SA, and HA-SA, were sterilized by autoclaving or UV irradiation. We used a UV-C lamp with a wavelength of 254 nm. Samples were exposed to UV-C light for 30 min on each side to ensure comprehensive coverage. The distance between the UV lamp and the samples was maintained at 10 cm, and the UV sterilization process was conducted in a sterile laminar flow hood to prevent contamination [15]. To ensure the validation of UV sterilization for our experiments the Direct Agar Contact Method was employed. After sterilization, the samples were incubated in Dulbecco’s Modified Eagle Medium (DMEM) supplemented with 10% fetal bovine serum (FBS) (Gibco, Thermo Fisher Scientific, Waltham, MA, USA) at 37 °C for 24 h to prepare sample eluates. These extracts were then used for treating the mesenchymal stem cells (MSCs) (Lonza (Basel, Switzerland), catalog number PT-2501), selected for the cytotoxicity testing. The cells were seeded into 96-well plates at a density of 1 × 10^4^ cells per well. This density was calculated by counting the viable cells using an Improved Neubauer hemocytometer (Hausser Scientific, Horsham, PA, USA) and then diluting the cell suspension to the desired concentration. The cells were allowed to adhere and grow overnight in an incubator set at 37 °C with 5% CO_2_. After the MSCs adhered and grew overnight, the liquid content in each well was replaced with 100 µL of the prepared sample extract. Each treatment was performed in triplicate to ensure the accuracy and reproducibility of the results. Control wells were included: negative controls containing only cell culture medium to represent normal cell growth, and positive controls with a known cytotoxic agent to validate the assay’s effectiveness. The cells were then incubated with the sample extracts for 24 h under the same conditions. Following the 24 h incubation, the MTT (3-(4,5-dimethylthiazol-2-yl)-2,5-diphenyltetrazolium bromide) assay was conducted to assess cell viability. The medium was removed from each well, and 100 µL of fresh culture medium containing 0.5 mg/mL of MTT reagent was added to each well. The plates were then incubated for an additional 4 h at 37 °C, allowing the MTT reagent to be metabolized by viable cells, forming formazan crystals. After incubation, the MTT solution was carefully aspirated, and 100 µL of dimethyl sulfoxide (DMSO) was added to each well to dissolve the formazan crystals. The absorbance of each well was then measured at 570 nm using a plate reader. To convert the absorbance readings into cells/mL, a standard curve was generated, creating a range of known MCS concentrations. The absorbance values were then plotted against the corresponding cell concentrations to create a standard curve. A linear regression analysis was performed to derive the equation of the line (Absorbance = m × Cell Concentration + b), which was used to convert the absorbance readings from the experimental wells into the number of viable cells per milliliter. The absorbance values correlated with the number of viable cells, allowing for the assessment of cytotoxicity. The calculated cell viability percentages were then compared to determine the cytotoxicity of each sample. Low cytotoxicity was indicated by cell viability greater than 80%, moderate cytotoxicity by viability between 50% and 80%, and high cytotoxicity by viability less than 50% [16]. To ensure the accuracy and reliability of the results, ISO 10993-5:2009 standards were followed for this study (Figure 3).

### 2.5. Testing the Antibacterial Activity

#### 2.5.1. Experimental Design

We tested the antibacterial effect of the samples in solid state as tablets and as coatings (on the surface of Ti structures) using the MAPLE technique. The studies aimed to determine whether the CPX compound retained its therapeutic potential in different pharmaceutical forms. To establish whether the inclusion of CPX in the HA structure modified its antibacterial activity, we also prepared a mechanical mixture based on HA and CPX. This evaluation is crucial because bone infections such as osteomyelitis can be challenging to treat due to the difficulty of delivering effective antibiotic concentrations directly to the site of infection. By incorporating CPX into HA and Ti-HA matrices, we aimed to enhance local antibiotic delivery, reduce systemic side effects, and improve overall treatment outcomes.

The following tablets and coatings on the surface of Ti structures were prepared and analyzed:–HA-coated Ti substrates in tablet form (Ti-HA) by MAPLE: 6 micrograms of HA.–HACPX_CS_-coated Ti substrates in tablet form (Ti-HA-CPX) by MAPLE: 1 microgram of HA and 5 micrograms of CPX.–tablets of HACPX_CS_ composite obtained by chemical synthesis: 20 micrograms of HA and 5 micrograms of CPX.–tablets made from a mechanical mixture of CPX and HA (HACPX_MM_): 20 micrograms of HA and 5 micrograms of CPX.–tablets from synthetic HA (HA-synthesis): 25 micrograms.–tablets of standard CPX (Sigma-Aldrich)-CPX-SA: 5 micrograms.–tablets of standard HA from Sigma-Aldrich, HA-SA: 25 micrograms (Figure 3).

#### 2.5.2. Preparation of the Culture Medium and Inoculation

Mueller-Hinton nutrient agar (Oxoid, Thermo Fisher Scientific, USA) was the standard medium utilized for disc diffusion antimicrobial susceptibility testing, being distributed into Petri dishes with a diameter of 100 mm, forming a uniform layer of 4 mm. Agar plates and bacterial lawns were prepared according to standard protocols and referenced in ISO 20776-2:2021. The inoculum was prepared by suspending colonies in saline to achieve a final concentration of approximately 10^8^ CFU/mL. The turbidity of the suspension was measured nephelometrically to match a 0.5 McFarland standard. The culture medium was maintained at a pH of 7.2–7.4, ensuring an appropriate composition for the optimal growth of the bacterial species under investigation. The application involved flooding the nutrient medium with the bacterial suspension, followed by the removal of excess fluid. The inoculated plates were then dried at room temperature (22 °C) for 10 min before sample application. The microorganisms tested were derived from standard reference strains obtained from the Cantacuzino Institute, which are known to be sensitive to the selected antibiotic (CPX).

#### 2.5.3. Determination of the Minimum Inhibitory Concentration of CPX

To determine the Minimum Inhibitory Concentration (MIC), filter paper discs impregnated with a standard gradient of concentrations of CPX (2, 3, 4, 5, 6, 7, 8 µg) were placed onto culture media inoculated with *Staphylococcus aureus* (ATCC 25923) and *Escherichia coli* (ATCC 25922). The Petri dishes were then incubated at 37 °C for 24 h. The MIC was defined as the lowest concentration of the antibiotic that completely inhibits the visible growth of bacteria in vitro (Table 1).

#### 2.5.4. Performing Antibiogram Testing

To test the antibacterial effect of CPX and other samples (Ti-HA, Ti-HA-CPX, HACPX_CS_, HACPX_MM_, HA-synthesis, CPX-SA, and HA-SA), we employed the disc diffusion method on nutrient agar (Kirby-Bauer), in accordance with pharmacopoeia FR X. Approximately 15 min after inoculation, samples were deposited onto the surface of the culture medium using ophthalmic forceps. Each sample (tablets of 6 mm diameter) was placed 1.5 cm from the edge of the Petri dish and 3 cm apart from each other. The Petri dishes were incubated inverted for 18 h at 37 °C. Results were measured using a graduated ruler, noting the diameter of the inhibition zone in millimeters. The results were expressed as average values calculated from the arithmetic mean of three measurements. Very small colonies, subsequent invasion of the inhibition zone, and discrete growths within the inhibition zone were not considered. The final results were categorized as sensitive (S), intermediate sensitivity (IS), and resistant (R) [17].

### 2.6. Statistical Analysis

The statistical analysis was conducted to evaluate the differences in antibacterial efficacy and cell viability among various bioceramic treatments. The analysis included one-way Analysis of Variance (ANOVA), followed by post-hoc Tukey’s Honestly Significant Difference (HSD) tests and pairwise t-tests. These statistical methods were employed to identify significant differences between the groups and to ensure robust comparative analysis.

A one-way ANOVA was used to determine if there were statistically significant differences in the mean diameters of bacterial inhibition zones and cell viability percentages among the different treatments. The null hypothesis for the ANOVA was that there were no differences in means among the groups. A *p*-value of less than 0.05 was considered statistically significant. Following the ANOVA, post-hoc comparisons were made using Tukey’s HSD test to identify which specific groups differed from each other. Tukey’s HSD test controls the Type I error rate and is suitable for making multiple comparisons between group means.

In addition to the ANOVA and Tukey’s HSD tests, pairwise t-tests were conducted to further explore the differences between specific groups. These t-tests provided a detailed understanding of the pairwise comparisons, highlighting significant differences in antibacterial activity and cell viability between the treatments.

## 3. Results

For HA-CPX tablets prepared by mechanical mixing and pressing, the release profile was rapid, with 85% of CPX released within the first 3 days. The Higuchi model showed a higher correlation coefficient (R^2^ = 0.987) compared to the Ritger-Peppas model (R^2^ = 0.975), indicating that diffusion was the main release mechanism. For HA-CPX tablets synthesized chemically, 50% of CPX was released within the first 7 days, with 98% released by the 11th day. The higher correlation coefficient for the Higuchi model (R^2^ = 0.980) compared to the Ritger-Peppas model (R^2^ = 0.972) also suggested diffusion as the main release process. For Ti-HA-CPX coatings on titanium plates, the initial release was higher due to desorption from the HA surface, followed by slower diffusion from the HA matrix. The Higuchi model showed a correlation coefficient of 0.988, while the Ritger-Peppas model showed a slightly higher correlation coefficient of 0.992, suggesting that alongside diffusion, other processes such as surface erosion and desorption also contributed to the drug release (Figure 4).

The results show that the heat of formation for CPX, HACPX_CS_, and Ti-HA-CPX-CPX is −98.52 kcal/mol, −395.03 kcal/mol, and −514.53 kcal/mol, respectively. The increased stability (more negative values) of HACPX_CS_ and Ti-HA-CPX correlates with their enhanced antibacterial activity compared to CPX alone, as evidenced by larger inhibition zones in antibacterial tests (Ti-HA-CPX had an inhibition zone diameter of 33.5 mm for *Staphylococcus aureus* (ATCC 25923)) (Table 2). In the study, the dipole moment values are 7.85 Debye for CPX, 15.22 Debye for HACPX_CS_, and 10.96 Debye for Ti-HA-CPX. The higher dipole moment of HACPX_CS_ suggests greater reactivity, which can enhance its ability to disrupt bacterial cell membranes, contributing to its higher antibacterial activity. The total energy values in the study are −945.24 kcal/mol for CPX, −1968.04 kcal/mol for HACPX_CS_, and −2025.25 kcal/mol for Ti-HA-CPX. The significantly lower total energy of HACPX_CS_ and Ti-HA-CPX indicates greater stability and is associated with better antibacterial performance compared to CPX alone (Table 2).

In Table 3, the frontier orbitals of the CPX, HACPX_CS_, and Ti-HA-CPX are presented. The investigation of molecular surface areas for the two types of molecular orbitals reveals the contribution of atomic orbitals to their formation. Molecules with the highest E_HOMO_ values are most susceptible to electrophilic attack, while those with the highest E_LUMO_ values are more susceptible to nucleophilic attack. Based on the values provided in Table 2, the most stable molecule is Ti-HA-CPX, followed by HACPX_CS_. Table 4 provides the following ΔE values: 11.13 eV for CPX, 7.64 eV for HACPX_CS_, and 5.72 eV for Ti-HA-CPX. The smaller ΔE for HACPX_CS_ and Ti-HA-CPX correlates with their enhanced antibacterial activity, as these compounds are more reactive and capable of interacting more effectively with bacterial cells (Table 4).

Molecular docking studies show the binding modes and energies of CPX, HACPX_CS_, and Ti-HA-CPX with bacterial proteins. The binding energies are −4.68 ± 0.01 kcal/mol for *Escherichia coli* and −5.46 ± 0.03 kcal/mol for *Staphylococcus aureus*.

Ti-HACPX_CS_ shows the highest antibacterial efficacy against both *Staphylococcus aureus* (ATCC 25923) (33.5 ± 0.3 mm) and *Escherichia coli* (ATCC 25922) (27.5 ± 0.4 mm), indicating a strong synergistic effect of CPX when combined with Ti-HA. HACPX_CS_ and HACPX_MM_ show similar inhibition zones, suggesting that both methods of combining HA with CPX are effective. Both treatments exhibit high antibacterial activity, with inhibition zones indicating sensitivity to both bacterial strains. The standard CPX (CPX-SA) also shows high antibacterial activity, with inhibition zones of 29 ± 0.4 mm for *Staphylococcus aureus* (ATCC 25923) and 27.5 ± 0.3 mm for *Escherichia coli* (ATCC 25922), reinforcing CPX’s effectiveness as an antibiotic. HA-synthesis and HA-SA show much lower antibacterial activity, with HA-SA being particularly ineffective against *Staphylococcus aureus* (ATCC 25923) (14.5 ± 0.2 mm, resistant) and only intermediate against *Escherichia coli* (ATCC 25922) (17 ± 0.2 mm). This underscores the fact that HA has limited antibacterial properties on its own. The Ti-HA composite shows intermediate antibacterial activity, better than HA alone but significantly less effective than Ti-HA-CPX. This indicates that the antibacterial properties are primarily due to CPX’s incorporation. The molecular docking results in Table 5 align well with the antibacterial efficacy results in Table 6. Strong binding energies for CPX, HACPX_CS_, and Ti-HA-CPX correspond to high antibacterial activity, confirming the effectiveness of these composites in inhibiting bacterial growth. The superior performance of Ti-HA-CPX highlights the synergistic benefits of combining CPX with titanium and HA, leading to enhanced stability and release of the antibiotic, thereby improving its antibacterial efficacy.

Ti-HA, HA-synthesis, and HA-SA samples showed the highest cell viability at 95%, indicating that they are highly biocompatible with MSCs. Ti-HA-CPX, HACPX_CS_, and HACPX_MM_ exhibited moderate cytotoxicity with cell viability in the range of 75% to 80%. The presence of CPX in these composites likely contributed to the observed cytotoxic effects, as antibiotics can affect cellular metabolism and viability. The CPX-SA sample demonstrated the lowest cell viability at 50%, indicating high cytotoxicity. Direct exposure to pure CPX significantly affected the MSCs, likely due to the antibiotic’s potent activity interfering with cellular functions (Table 7).

The Tukey HSD test reveals that Ti-HA-CPX has significantly larger inhibition zones compared to other products, including HACPX_CS_, HACPX_MM_, and CPX-SA, indicating superior antibacterial efficacy against *Staphylococcus aureus* (ATCC 25923). Similar results were observed, with Ti-HA-CPX showing significantly higher antibacterial activity compared to other samples against *Escherichia coli* (ATCC 25922). For both *Staphylococcus aureus* (ATCC 25923) and *Escherichia coli* (ATCC 25922), Ti-HA-CPX and CPX-SA consistently show significantly higher antibacterial activity compared to other products (*p* < 0.001). For the cytotoxicity test, the Tukey HSD test shows that Ti-HA, HA-synthesis, and HA-SA have significantly higher cell viability compared to Ti-HA-CPX and CPX-SA. Ti-HA-CPX and CPX-SA exhibit lower cell viability, indicating higher cytotoxicity. Pairwise t-tests confirm the significant differences observed in the Tukey HSD test. Products such as Ti-HA-CPX and CPX-SA show significant differences in cell viability compared to other products, suggesting higher cytotoxicity. Ti-HA, HA-synthesis, and HA-SA are highly biocompatible, showing high cell viability and low cytotoxicity. Ti-HA-CPX and CPX-SA exhibit moderate to high cytotoxicity, which might limit their use in certain clinical applications where cell compatibility is crucial. The results suggest a trade-off between antibacterial efficacy and cytotoxicity. While Ti-HA-CPX is highly effective against bacteria, its higher cytotoxicity needs to be carefully managed.

## 4. Discussion

The enthalpy of formation is a measure of the heat energy released or absorbed when a chemical substance is formed; a low value indicates that the substance is more stable. The electric dipole moment reflects the partial separation of electric charge within a molecule, with a high value indicating increased reactivity [18,19,20]. The total energy expresses the stability of a molecule, with a low value of this descriptor being correlated with increased stability. By calculating the electrostatic potential, a three-dimensional map of the electron density was obtained, with electrostatic potential values indicated by different colors. The drug-substance-to-protein-target interaction, which produces the corresponding therapeutic effect, is only possible for a certain form of the molecule and a specific distribution of electron density within the molecule. This distribution generates regions rich in electrons (Table 3). The electrostatic potential can be used to assess molecular lipophilicity, as it characterizes the polarity of a specific region on the van der Waals surface of the molecule. Regions with high potential values will strongly attract polar molecules, while regions with low values will not and can be considered hydrophobic [21]. The presence of atoms with partial negative charges (such as halogens, sulfur, nitrogen, and oxygen) in the ligand molecule results in a large number of ligand-receptor interactions [22]. The value of the E_HOMO_ descriptor indicates the donor properties of a molecule, that is, its tendencies towards oxidation, while the value of the E_LUMO_ descriptor allows for the estimation of a molecule’s acceptor properties, that is, its reduction tendencies. The energy difference between the HOMO and LUMO levels (ΔE = E_LUMO_-E_HOMO_) is an important chemico-biological molecular descriptor, as it explains the stability of the molecule, with a low value indicating that the molecule has increased reactivity.

Research studies have reported that the HOMO-LUMO energy gap (ΔE) is an important indicator of antibacterial activity. In a study by Obot et al., it was shown that a smaller HOMO-LUMO gap often correlates with higher antibacterial activity, as it suggests greater chemical reactivity and potential for interaction with bacterial cells. Their research demonstrated that molecules with smaller ΔE values had better antibacterial efficacy, as confirmed through both in vitro and docking studies [23].

Other researchers have found that the heat of formation of a molecule can correlate with its antibacterial activity. For instance, a study by Chen et al. demonstrated that molecules with lower heat of formation values tend to exhibit higher antibacterial activity. This is because a lower heat of formation often indicates greater stability of the molecule, which may enhance its interaction with bacterial targets [24].

The dipole moment is an expression of the molecular system’s polarization and serves as a predictor of the chemical reactivity of the molecules, which affects their reactivity and interaction with bacterial cell membranes. Other researchers have shown that a higher dipole moment can improve the binding affinity of molecules in docking studies. This improved binding affinity is often linked to better antibacterial activity as stronger interactions with bacterial proteins are facilitated. Ishihara et al. highlighted that molecules with higher dipole moments showed enhanced docking scores against bacterial enzymes, correlating with their observed antibacterial efficacy [25].

Other researchers have observed a correlation between the total energy of a molecule and its antibacterial properties. Shah et al. found that molecules with lower total energy values tend to exhibit higher antibacterial activity. This is attributed to the fact that lower total energy reflects greater molecular stability and potentially more effective interaction with bacterial cell components [26].

Molecular docking results suggest that HACPX_CS_ and Ti-HA-CPX have strong binding affinities, which likely contribute to their higher antibacterial activities observed in vitro. The molecular docking study highlights the binding mode and energy of CXP, HACPX_CS_, and Ti-HA-CPX as being almost equal to those of the pure antibiotic, as detailed in Table 5. The 2D diagrams illustrate the types of bonds established between CPX and the protein targets (Figure 5 and Figure 6). These correlations demonstrate that more stable, reactive, and effectively binding molecules, as indicated by their molecular descriptors and docking studies, tend to exhibit enhanced antibacterial activity.

The sustained release profile of CPX from HACPX_CS_ and Ti-HA-CPX composites, maintaining prolonged antibacterial activity, is particularly significant for clinical applications. This feature could reduce the need for repeated antibiotic administration and minimize systemic side effects, which is crucial for treating chronic osteomyelitis and infected fractures. The ability of these composites to provide localized antibiotic delivery directly to the infection site offers a promising approach for single-stage surgical interventions, potentially improving treatment outcomes and reducing healthcare costs associated with prolonged antibiotic therapy and multiple surgeries.

Although the antimicrobial activity of CPX against the germs studied in this experiment is known, the binding of this compound to HA could modify the antimicrobial activity of the chemical compound [27]. Following the performance of antibiograms to determine the MIC, we have established that a dose of 5 µg CPX is the lowest amount of antibiotic that inhibits bacterial growth on the tested germs. Data on the antimicrobial activity of HA are quite contradictory. While initial studies on the chemical synthesis of HA did not refer to its antibacterial activity, using it as a negative control, more recently, there has been an increasing notion that it may have a weak antibacterial effect. A research group found that HA nanoparticles exhibit significant antibacterial activity against both Gram-positive and Gram-negative bacteria, suggesting their potential application in biomedical devices to prevent infections. This study demonstrated that the antibacterial properties were due to the physical disruption of bacterial cell membranes and the release of calcium and phosphate ions, which interfered with bacterial metabolism [28]. In another study, researchers investigated the synergistic effects of HA and various antibiotics. They found that the combination of HA with antibiotics such as tetracycline and vancomycin significantly enhanced antibacterial efficacy compared to antibiotics alone. The study concluded that HA could serve as an effective carrier to enhance the delivery and potency of antibiotics [29]. Another investigation revealed that silver-doped HA exhibits enhanced antibacterial activity against multi-drug-resistant bacterial strains. The incorporation of silver ions into the HA matrix provided a sustained release of silver, which effectively inhibited bacterial growth and biofilm formation. This study highlights the potential of silver-doped HA as a multifunctional material for use in bone implants to reduce the risk of post-surgical infections [30]. In another study, a vancomycin-loaded HA/collagen/PLA bone graft substitute was developed that exhibited a high bacterial inhibition rate and strong adherence to the damaged site without causing inflammatory reactions. This study highlights the infection-inhibition properties of HA composites when combined with antibiotics, supporting their use in preventing infections in orthopedic applications [31]. As observed in the results presented in Table 6, HA deposits on substrates are characterized by a weak antibacterial effect. The inhibition zone diameters for HA samples used in the study are approximately the same, with higher values in the case of *Escherichia coli* (ATCC 25922). This is consistent with literature data indicating that HA nanoparticles exhibit a more pronounced antibacterial character against *Escherichia coli* (ATCC 25922), explained by the relatively thin cell wall thickness characteristic of this bacterial species and its specific chemical structure. Although the antimicrobial activity of CPX against the studied bacteria is well known, binding this compound to HA could potentially modify the antimicrobial activity of the chemically synthesized compound.

Our results with HACPX_CS_ and HACPX_MM_ showed similar antibacterial activities, with inhibition zones of 26 mm for both *Staphylococcus aureus* (ATCC 25923) and *Escherichia coli* (ATCC 25922). This suggests that, like gentamicin in Suchý et al.’s study, CPX retains its antibacterial efficacy when combined with HA. However, the 30% reduction seen with vancomycin in Suchý et al.’s study is not observed in our results, indicating a stable integration of CPX within the HA matrix [32]. Gomes et al.’s findings support our observations of sustained antibacterial activity in HACPX_CS_ composites. Both HACPX_CS_ and HACPX_MM_ demonstrated substantial antibacterial zones, indicating effective and sustained release of CPX akin to the amoxicillin release observed in Gomes et al.’s study [33]. Prasanna et al. found that the antimicrobial properties of CPX and tetracycline embedded in silver-doped HA (AgHA-C and AgHA-T) suspensions were tested against *Staphylococcus aureus* (ATCC 25923), *Escherichia coli* (ATCC 25922), and *Candida albicans*. The results showed that both AgHA-C and AgHA-T suspensions exhibited exceptional antibacterial activity, which increased with longer incubation times. Our study did not observe a significant decrease in antibacterial activity in HACPX_CS_ composites compared to CPX alone. This suggests that the interaction between CPX and HA in our composites may be more stable, preserving CPX’s efficacy [34]. The high antibacterial activity of Ti-HA-CPX in our study, with inhibition zones of 33.5 mm for *Staphylococcus aureus* (ATCC 25923) and 27.5 mm for *Escherichia coli* (ATCC 25922), aligns well with Predoi et al.’s findings. The integration of CPX in HA on Ti substrates enhances antibacterial efficacy, similar to the vancomycin-coated Ti implants in their study [35]. Our HACPX_CS_ composites’ effective antibacterial activity and sustained release are consistent with Amarnath et al.’s findings. The retention of antibacterial properties in our HACPX_CS_ samples mirrors the sustained efficacy of rifampicin in HA-gelatin composites. *Staphylococcus aureus* (ATCC 25923) [36]. The prolonged antibacterial activity of Ti-HA-CPX in our study aligns with Wang et al.’s findings. The effective integration of CPX in HA and Ti substrates supports its potential for treating chronic infections, similar to levofloxacin in HA beads. The moderate cytotoxicity observed in our HACPX_CS_ composites (75–80% cell viability) is consistent with other researchers findings of acceptable cytotoxicity levels in HA-antibiotic biocomposites. This suggests that HA can effectively control antibiotic release while maintaining biocompatibility [37]. The sustained antibacterial activity observed in our HACPX_CS_ composites parallels Alvarez et al.’s findings, indicating that HA matrices can effectively manage antibiotic release and maintain antimicrobial efficacy [38]. Unlike Shinde et al.’s findings, our study did not observe a significant reduction in CPX’s antibacterial activity when incorporated into HA. This indicates a more stable integration of CPX in HA, preserving its efficacy [39]. Our results differed from Ghosh et al.’s observations as CPX maintained its antibacterial efficacy when combined with HA, suggesting a stable incorporation that prevents significant loss of activity [40]. Therefore, we prepared samples in the form of tablets, obtained both from the chemically synthesized composite and the mechanical mixture of HA and CPX, maintaining the same ratio as in our synthesized compound (20%). The antibiogram results indicate that binding CPX to HA minimally modifies antibacterial activity (inhibition zone diameter decreases by less than 2% upon binding to HA). An aspect observed in this study is a slight increase in the inhibition zone when the studied compounds are deposited on Ti substrates, compared to when they are in tablet form. It is well known that Ti-based alloys are covered by a spontaneously forming TiO_2_ layer due to oxidation reactions under light influence.

Several studies in the literature show that this oxide layer has antimicrobial properties. Given that the film deposited through MAPLE is found only on one side of the Ti discs, the other side (which contains TiO_2_) could enhance the antimicrobial effect [41].

Another study shows significant limitations regarding the use of the antibacterial effect of the TiO_2_ layer in biomedical applications, indicating that this layer must be designed according to the type of bacteria, their attachment to the surface, their growth mechanism, and the environment in which they are found [42].

The antibacterial efficacy of bioceramic samples, as indicated by larger inhibition zones (Table 6), must be balanced against their cytotoxic effects (Table 7). For instance, Ti-HA-CPX showed the highest antibacterial activity but moderate cytotoxicity, suggesting its potential for clinical use where infection prevention is paramount, but careful consideration of dosage and release mechanisms is necessary to minimize cytotoxic effects. The incorporation of antibiotics such as CPX into bioceramic matrices should be optimized to enhance antibacterial efficacy while maintaining acceptable cytotoxicity levels. Techniques such as controlled release, surface modifications, and composite formulations can help achieve this balance. Our study showed that Ti-HA, HA-synthesis, and HA-SA exhibited low cytotoxicity with a cell viability of 95%. This high viability suggests that these bioceramics do not significantly affect cell growth and proliferation, making them suitable for applications where biocompatibility is crucial, such as in bone regeneration and repair. This high viability indicates excellent biocompatibility, consistent with the findings of Pupo et al., who reported that HA scaffolds without antibiotics supported high cell viability and were suitable for bone regeneration applications. Similarly, Song et al. found that HA-gelatin composites without antibiotics showed high biocompatibility, supporting cell adhesion and proliferation [43].

Ti-HA-CPX, HACPX_CS_, and HACPX_MM_ samples demonstrated moderate cytotoxicity, with cell viability ranging from 75% to 80%. This is in line with the findings of Patel et al., who observed that HA-alginate biocomposites loaded with clindamycin showed moderate cytotoxic effects but remained within an acceptable range for clinical applications. Additionally, Zheng et al. reported that HA-chitosan composites loaded with doxycycline exhibited moderate cytotoxicity while maintaining effective antibacterial properties. CPX alone exhibited high cytotoxicity, with cell viability at 50%. This is consistent with the observations by Shinde et al., who reported significant cytotoxic effects of cefazolin when not embedded in a biocompatible matrix, reducing cell viability substantially. Kim et al. also found that azithromycin combined with HA showed reduced cell viability due to the adsorption properties of HA, which affected the antibiotic’s bioavailability and increased cytotoxicity [30,44,45].

The present study is limited by a relatively small sample size, which impacts the generalizability and statistical power of the findings. Future research with larger sample sizes is necessary to validate these findings and provide more comprehensive insights into the antibacterial efficacy and cytotoxicity of CPX-functionalized bioceramic implants. Expanding the sample size would enhance the reliability of the results and improve the applicability of the study’s conclusions to a broader context.

## 5. Conclusions

This study highlights the significant potential of HACPX_CS_ and Ti-HA-CPX composites in enhancing antibacterial efficacy. These composites exhibited greater stability and superior antibacterial activity compared to CPX alone, as shown by larger bacterial inhibition zones. The higher dipole moments of HACPX_CS_ and Ti-HA-CPX indicate increased chemical reactivity, contributing to effective bacterial membrane disruption and improved antibacterial performance. Molecular docking studies confirmed strong binding affinities with bacterial proteins, further supporting their enhanced antibacterial efficacy. Additionally, the sustained release profile of CPX from these composites maintained prolonged antibacterial activity, potentially reducing the need for repeated antibiotic administration and minimizing systemic side effects. The functionalization of bioceramic implants with antibiotics offers promising advantages for single-stage surgical interventions, enhancing treatment outcomes for chronic osteomyelitis and infected fractures. In summary, antibiotic-functionalized bioceramic implants present a promising advancement in bone infection treatment, warranting further clinical studies to fully explore their potential.

## Figures and Tables

**Figure 1 pharmaceutics-16-00998-f001:**
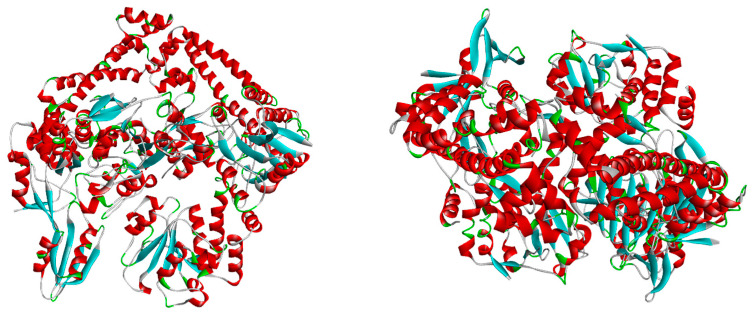
X-ray targets of *Staphylococcus aureus* (**left**) and *Escherichia coli* (**right**) gyrase.

**Figure 2 pharmaceutics-16-00998-f002:**
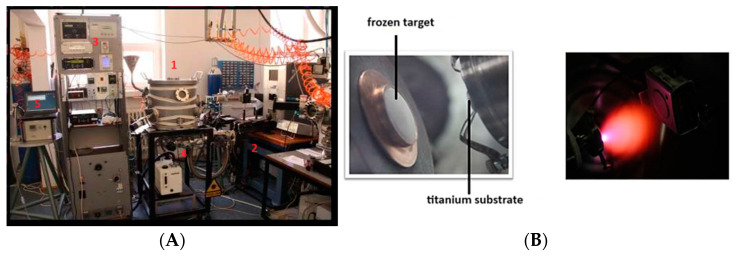
Image of the setup used in the Laser-Surface-Plasma Interactions (LSPI) Laboratory, Laser Department, National Institute for Laser, Plasma, and Radiation Physics, Bucharest, for MAPLE depositions. 1. Vacuum chamber; 2. KrF Excimer laser; 3. Monitoring device rack; 4. Vacuum pump and 5. PC data acquisition system (**A**). Image from the irradiation chamber. The target obtained by cryogenizing the HACPX_CS_ and solvent mixture is irradiated with a UV laser. The compound is transferred to a titanium substrate located 3–4 cm from the target (**B**).

**Figure 3 pharmaceutics-16-00998-f003:**
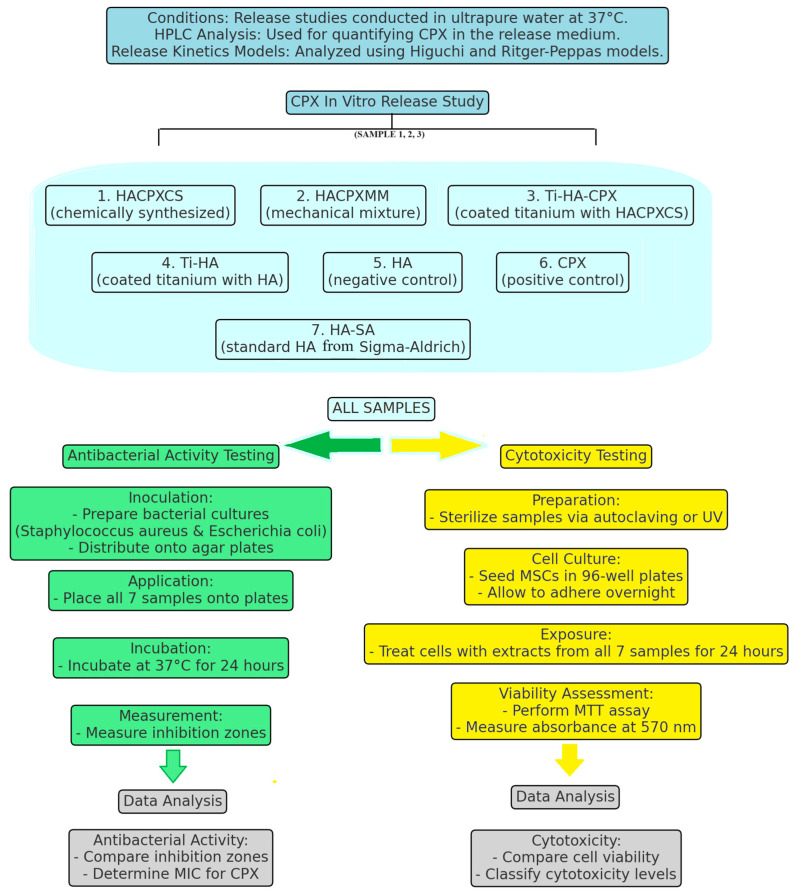
Diagram of the citotoxicity test and antibacterial activity testing procedure.

**Figure 4 pharmaceutics-16-00998-f004:**
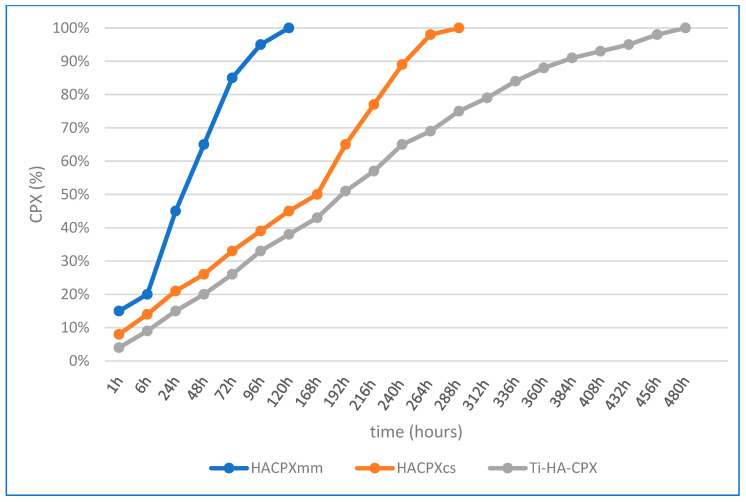
The percentage of CPX released from the test samples over time.

**Figure 5 pharmaceutics-16-00998-f005:**
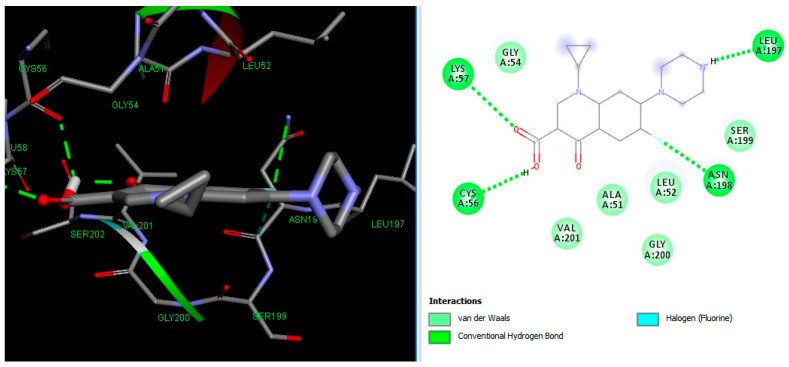
3D image (**left**) and 2D interaction map (**right**) of CPX-*Escherichia coli*. Amino acids in the active site are depicted as spheres, with the antibiotic positioned centrally on the map. Dotted lines demonstrate the interaction mode between the ligand and the target.

**Figure 6 pharmaceutics-16-00998-f006:**
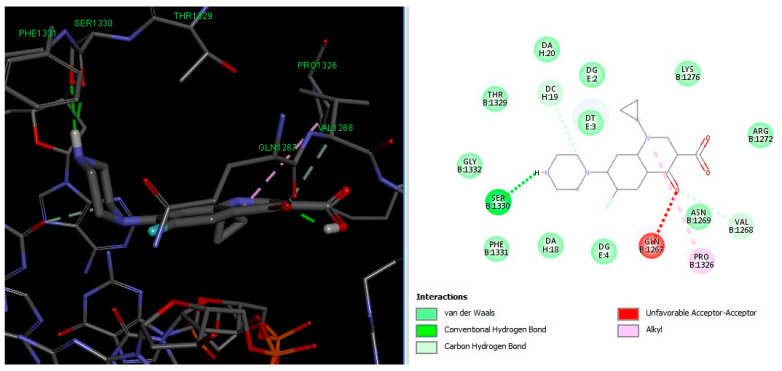
3D image (**left**) and the 2D interaction map (**right**) of the CPX-*Staphylococcus aureus* interaction. The amino acids in the active site are represented as spheres, with the antibiotic located at the center of the map. Dotted lines indicate the mode of interaction between the ligand and the target.

**Table 1 pharmaceutics-16-00998-t001:** The standardized criteria for interpreting the diameters of the inhibition zones according to ISO 20776-1:2019.

Microorganism Test	R (mm)	IS (mm)	S (mm)
*Staphylococcus aureus* (ATCC 25923)	≤15	16–20	≥21
*Escherichia coli* (ATCC 25922)	≤15	16–20	≥21

**Table 2 pharmaceutics-16-00998-t002:** Value of Important CPX Descriptors and CPX Release Structures.

Sample	Heat of Formation (kcal/mol)	Dipole Moment (Debye)	Total Energy (kcal/mol)
CPX	−98.52	7.85	−945.24
HACPX_CS_	−395.03	15.22	−1968.04
Ti-HA-CPX	−514.53	10.96	−2025.25

**Table 3 pharmaceutics-16-00998-t003:** The electrostatic potential and the frontier orbitals HOMO and LUMO for the ligands used. In all images, the green color indicates a positive polarity, while the violet color indicates a negative polarity.

Sample	Electrostatic Potential	HOMO Orbitals	LUMO Orbitals
CPX	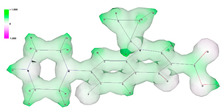	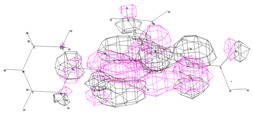	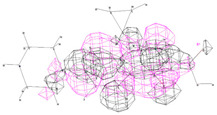
HACPX_CS_	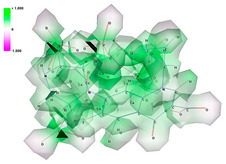	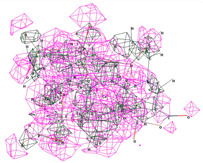	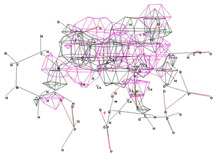
Ti-HA-CPX	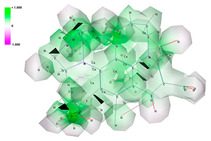	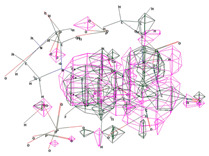	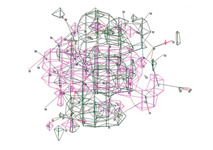

**Table 4 pharmaceutics-16-00998-t004:** The Frontier Orbital Values of HOMO and LUMO.

Sample	HOMO (eV)	LUMO (eV)	ΔE
CPX	−9.3214	1.8151	11.13
HACPX_CS_	−8.3535	−0.7128	7.64
Ti-HA-CPX	−7.3649	−1.6377	5.72

**Table 5 pharmaceutics-16-00998-t005:** Ligand-target binding energy (kcal/mol).

Ligands	*Escherichia coli*	*Staphylococcus aureus*
CPX/HACPX_CS_/Ti-HA-CPX	−4.68 ± 0.01	−5.46 ± 0.03

**Table 6 pharmaceutics-16-00998-t006:** Average Diameter of Bacterial Growth Inhibition. Determining the antibacterial effect of tested samples.

**DZI** **mm**	**Product Tested**	** *Staphylococcus aureus* **	** *Escherichia coli* **
	Ti-HA	17.5 ± 0.5 **	18 ± 0.5 **
	Ti-HA-CPX	33.5 ± 0.3 ***	27.5 ± 0.4 ***
	HACPX_CS_	26 ± 0.2 ***	26± 0.3 ***
	HACPX_MM_	26.5 ± 0.2 ***	26± 0.2 ***
	HA-synthesis	15 ± 0.3 *	17.5± 0.4 **
	CPX- SA	29 ± 0.4 ***	27.5± 0.3 ***
	HA-SA	14.5 0.3 *	17± 0.2 **

DZI-A is the average value of the diameters of bacterial inhibition zones (mm). * resistant; ** intermediate; *** sensitive.

**Table 7 pharmaceutics-16-00998-t007:** Cytotoxicity evaluation of bioceramic samples using mesenchymal stem cells.

Product Tested	Cell Viability (%)	Cytotoxicity Level
Ti-HA	95%	Low cytotoxicity
Ti-HA-CPX	80%	Moderate cytotoxicity
HACPX_CS_ (chemical synthesis)	75%	Moderate cytotoxicity
HACPX_MM_ (mechanical mixture)	75	Moderate cytotoxicity
HA-synthesis	95%	Low cytotoxicity
CPX-SA	50	High cytotoxicity
HA-SA	95%	Low cytotoxicity

## Data Availability

Data are contained within the article.

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
