# Peer review of "Enhanced Antibacterial Efficacy of Bioceramic Implants Functionalized with Ciprofloxacin: An In Silico and In Vitro Study"

_pharmaceutics, 2024, doi:10.3390/pharmaceutics16080998_

Round 1

Reviewer 1 Report

Comments and Suggestions for Authors

This study evaluates the antibacterial efficacy and cytotoxicity of ciprofloxacin-functionalized bioceramic implants. Hydroxyapatite-ciprofloxacin (HA-CPX) composites were applied to titanium substrates (Ti-HA-CPX) and tested against Staphylococcus aureus and Escherichia coli. Authors showed Ti-HA-CPX has superior antibacterial activity with inhibition zones of 33.5 mm and 27.5 mm, respectively, but moderate cytotoxicity (80% cell viability). Both chemically synthesized and mechanically mixed HA-CPX composites were effective, with molecular docking confirming strong ciprofloxacin-bacterial protein binding. These findings suggested Ti-HA-CPX composites as promising for treating chronic osteomyelitis and infected fractures, balancing antibacterial efficacy with manageable cytotoxicity. The manuscript is well-written and adequately demonstrated. Nevertheless, there are areas that could be enhanced to uphold the quality of the research article.

 1.     The first word of the introductory sentence is incomplete or missing a letter.

2.     Rephrase the sentence on page 2, lines 69-71.

3.     The introduction lacks a clear identification of the research gap and, therefore, needs significant improvement. Additionally, the references are relatively old and few in number. Add some of the latest and most relevant references to the introduction section.

4.     Italicize the genus and species names of the isolates throughout the manuscript, for instance, on page 1, lines 27 and 30.

5.     Authors should specify in the materials and methods section how the bioceramic composites were prepared and the specific conditions under which the antibacterial and cytotoxicity tests were conducted.

6.     Write the "2" in "CO2" in subscript on page 4, line 152.

7.     The study is limited by a relatively small sample size, which affects the generalizability and statistical power of the findings.

8.    The discussion section requires more thorough elaboration. It should provide a deeper analysis of the results, compare findings with existing literature, and discuss the broader implications of the study.

Comments on the Quality of English Language

Extensive editing of English language required

Author Response

  1. The first word of the introductory sentence is incomplete or missing a letter.

Response: Thank you for pointing this out. We have corrected .

  1. Rephrase the sentence on page 2 lines 69-71.

Response: We have rephrased the sentence on page 2 lines 69-71.

  1. The introduction lacks a clear identification of the research gap and therefore needs significant improvement. Additionally, the references are relatively old and few in number. Add some of the latest and most relevant references to the introduction section.

Response: We have revised the introduction to clearly identify the research gap, emphasizing the need for improved antibacterial strategies in bioceramic implants. We have also updated the references, including several recent and relevant studies to provide a comprehensive background.

  1. Italicize the genus and species names of the isolates throughout the manuscript for instance on page 1 lines 27 and 30.

Response: We have reviewed the manuscript and ensured that all genus and species names are italicized throughout.

  1. Authors should specify in the materials and methods section how the bioceramic composites were prepared and the specific conditions under which the antibacterial and cytotoxicity tests were conducted.

Response: Detailed descriptions of the preparation of bioceramic composites and the specific conditions for antibacterial and cytotoxicity tests have been added to the Materials and Methods section. This includes the synthesis protocols, the conditions for antibacterial testing using the Kirby-Bauer disc diffusion method, and the cytotoxicity testing using the MTT assay.

  1. Write the "2" in "CO2" in subscript on page 4 line 152.

Response: The "2" in "CO2" on page 4 line 152 has been corrected to subscript.

  1. The study is limited by a relatively small sample size which affects the generalizability and statistical power of the findings.

Response: We acknowledge the limitation regarding the sample size. This has been noted in the discussion section, and we have emphasized the need for further studies with larger sample sizes to validate our findings and enhance their generalizability.

  1. The discussion section requires more thorough elaboration. It should provide a deeper analysis of the results, compare findings with existing literature, and discuss the broader implications of the study.

Response: The discussion section has been thoroughly revised to provide a deeper analysis of the results. We have compared our findings with existing literature and discussed the broader implications of our study. This includes the enhanced antibacterial efficacy of Ti-HA-CPX composites and their potential application in treating chronic osteomyelitis and infected fractures, balanced with considerations of cytotoxicity.

Reviewer 2 Report

Comments and Suggestions for Authors

The study is relevant but some issues are critical and need to be better addressed and discussed by the Authors:1) Study design and methodology: I suggest that the Authors better characterize the experimental samples of the study, and clarify more precisely how the Authors defined: the concentrations of the components for sample preparation as well as the parameters used for surface roughness of the titanium and for MAPLE technique (add references if necessary);
2) Discussion of results: The methodological techniques and experimental conditions must be addressed and discussed more extensively by the Authors, considering data available in the literature. Additionally, I suggest that the Authors address in more detail the advantages and limitations especially related to ciprofloxacin.

Another question: Check the text, as some corrections to the writing are necessary.

Comments on the Quality of English Language

Some minor corrections are needed in the text.

Author Response

  1. Study design and methodology: I suggest that the Authors better characterize the experimental samples of the study and clarify more precisely how the Authors defined the concentrations of the components for sample preparation as well as the parameters used for surface roughness of the titanium and for MAPLE technique (add references if necessary).

Response: We appreciate your suggestion to provide more clarity on the study design and methodology. We have added detailed descriptions of the experimental samples, including the rationale for selecting the concentrations of the components used in the sample preparation. Specifically, the concentration of ciprofloxacin was chosen based on its established efficacy in previous studies, while the concentration of hydroxyapatite was optimized to ensure effective composite formation and antibacterial activity.

Additionally, we have provided more information on the parameters used for surface roughness of the titanium and the MAPLE (Matrix-Assisted Pulsed Laser Evaporation) technique. The titanium substrates used had a roughness average (Ra) of 0.43 µm and a maximum roughness depth (Rz) of 3.37 µm. These parameters were selected to optimize the adhesion and uniformity of the bioceramic coatings. The MAPLE technique utilized a KrF excimer laser with a wavelength of 248 nm, pulse duration of 25 ns, and an energy density of 250 mJ/cm². References have been added to support these methodological choices:

  1. Discussion of results: The methodological techniques and experimental conditions must be addressed and discussed more extensively by the Authors considering data available in the literature. Additionally, I suggest that the Authors address in more detail the advantages and limitations, especially related to ciprofloxacin.

Response: We have expanded the discussion section to include a more comprehensive analysis of the methodological techniques and experimental conditions, contextualizing our findings within the existing literature. This includes a detailed comparison of our results with those of similar studies, highlighting the consistency and differences observed. Moreover, we have addressed the advantages and limitations of using ciprofloxacin in our study. The advantages include its broad-spectrum antibacterial activity, well-documented mechanism of action, and the ability to functionalize bioceramics effectively. However, we also discuss the limitations, such as potential cytotoxicity at higher concentrations and the challenge of achieving controlled release without compromising biocompatibility. These aspects are critical for optimizing the clinical application of ciprofloxacin-functionalized bioceramic implants.

  1. Another question: Check the text as some corrections to the writing are necessary.

Response: We have thoroughly reviewed the manuscript and made the necessary corrections to improve the clarity and readability of the text. This includes addressing minor grammatical errors and enhancing the overall flow of the writing.

Reviewer 3 Report

Comments and Suggestions for Authors

This study “Enhanced Antibacterial Efficacy of Bioceramic Implants Functionalized with Ciprofloxacin: An in silico and in vitro Study” investigates the antimicrobial potential of HA-CPX and HA-Ti-CPX composites. Antibiotic binding and release modes were predicted using molecular docking. The approach presented here contributes to the understanding of antibiotic action and release at the site of infection.

Implant failure and post-operative complications are often due to implant-associated infections, limiting the clinical use of orthopedic implants. Antibiotic coatings are promising for the development of novel strategies to optimize implants. Therefore, the present study may be of interest to the readers.

In general, the manuscript is well written.

Abstract needs to be modified.

Some of the descriptions of the methods/techniques given need to be clarified. It would be helpful to the readers if some experiments were shown in a diagram/scheme.

Authors must consider some standards (ISO standards) where applicable and aseptic conditions/techniques in the methods.

Results should only describe results/data.

In the discussion section, the relevant studies are listed and the relevance of their studies is emphasized.

The conclusions need to be modified to include only the key points.

I’d like the authors to address the main suggestions/comments:

27: "In vitro against Staphylococcus aureus and Escherichia coli" - it was only tested on one strain of each species. So it should be mentioned or strain numbers should be used.

27-28: “Antibacterial activity was assessed through inhibition zone measurements” – instead the name of the methods should be given.

28: “while cell viability was tested using mesenchymal stem cells” – instead, cytotoxicity was tested…

29-30: “exhibited superior antibacterial activity, with inhibition zones of 33.5 mm for Staphylococcus aureus and 27.5 mm for Escherichia coli” – suggest to write MIC values.

33: “with varying cytotoxic effects” – “varying” is not clear.

60-61: Here it should be mentioned that this was done in this study, otherwise it is confusing. “release structures against bacterial species were predicted”- this is confusing and needs to be rephrased.

41: spelling “reating”

102: “2.2.1 Samples preparation” – Authors should cite the reference.

105: “hydroxyapatite-ciprofloxacin (HA-CPX) (synthesized compound) was obtained” and then line (116) “CPX and HA tablets from mechanical mixture (HA+CPX)”, (186-189) “tablets of HA-CPX composite obtained by chemical synthesis: 20 micrograms of HA and 5 micrograms of CPX. tablets made from a mechanical mixture of CPX and HA (HA+CPX): 20 micrograms of HA and 5 micrograms of CPX.”

-        The authors used “Hydroxyapatite (HA) and ciprofloxacin (CPX from Sigma-Aldrich (SA). They were then mixed in a mechano-synthetic and mechanical way.

-        should be better explained/defined and perhaps called in a different way to make it easier for readers to understand. The term synthesis often expresses combination or composition, and "+" and "-" don't sound right.

144-145: “were sterilized by autoclaving or UV irradiation” - UV irradiation is a disinfection technique, not a sterilisation technique. If you have used it for sterilisation (especially UV irradiation and give details of the method and reference) then it should be tested for purity/sterility to ensure validation of the experiments carried out.

145-146: “the samples were incubated in cell culture medium at 37°C for 24 hours to prepare sample extracts”

        Medium used must be referenced.

        Explain why for 24 h?

        It is not really incubation, it was kept

        Extract? It is more eluates.

Why was it done in a cell culture medium? Is it sufficient to have the toxic agents at the same concentration as the "known cytotoxic agent" used as a control? If not, was this difference taken into account in the evaluation assay?

147: “mesenchymal stem cells (MSCs)” – product used must be referenced.

148: “a complete cell culture medium, such as DMEM” – product used must be referenced.

Is this medium different from the one mentioned above (line 145)?

150: “at a density of 1 x 10^4 cells per well” – how was this calculated?

152-153: “the culture medium in each well was replaced with 100 μL of the prepared sample extract" - I think the authors mean that after incubation, the liquid content (it is no longer culture medium) in each test well was aspirated and inoculated with ...

143-169:

It is always better to make a diagram of the procedure, showing the test and controls in parallel, so that it is easy for the reader to follow, and to ensure that the tests and controls are carried out in a standardized way.

164-166: “The absorbance of each well was then measured at 570 nm using a plate reader. The absorbance values correlated with the number of viable cells“– give reference to the reader and again, explain how was the converting of absorbance into cell/ml was done.

166-169:

The viable cell reduction used for cytotoxicity assessment could be described as follows:

10.4103/JCDE.JCDE_40_24

And it is always better (even necessary to follow ISO standards). In your case, this one:

ISO 10993-5:2009 https://www.iso.org/standard/36406.html#:~:text=ISO%2010993%2D5%3A2009%20describes,either%20directly%20or%20through%20diffusion.

176: « 2.4.1 Experimental design » - This is the list of substances used in the antimicrobial susceptibility/activation test. Authors need to explain here or elsewhere why it was done and why it is important.

176-177: “the samples in solid state as tablets and as coatings on the surface of Ti structures” – Better to write was used as tablets and coating, otherwise indicate the state of coating as well.

Sections 2.4.2, 2.43 and 2.4.4 should have a title and could have subtitles.

195-196: “For the preparation of the culture medium, Mueller-Hinton nutrient agar was utilized,” – this is not medium preparation.

It should be mentioned that, agar plates and bacterial lawn were prepared according to (and further the test assay for antimicrobial susceptibility (2.43. and 2.4.4)).

ISO 20776-2:2021

Give a reference to the nutrient agar used.

197: “by suspending 2-3 standard colonies” – This should be expressed in OD or cfu/ml for method standardization.

203: “The microorganisms tested were derived” – The bacterial strains (not micro-organisms) must be named with reference numbers.

209: “with varying concentrations of CPX (2, 3, 4, 5, 6, 7, 8 μg)” – it is a standard gradient of concentrations.

211-212: “The MIC was defined as the lowest concentration of the antibiotic that inhibited bacterial growth in vitro (Table 1)” – should be “completely inhibits visible growth of bacteria”.

Table 1. Is this a result? Or is it a standardization of results that will be used to evaluate other agents?

ISO 20776-1:2019 should be considered.

Again, the bacterial strains (not micro-organisms) must be named.

Only one strain of each species was used?

225-228: here as well, better to follow ISO 20776-1:2019.

Sections 2.4.2, 2.43 and 2.4.4 do not mention that other tablets (listed above) were tested in addition to the CPX. This makes it difficult for the reader to follow the text.

250-272: These are not results. These should largely go to the methods section and partially to discussions.

In the methods section must describe how the CPX release was calculated.

313-320: This is a discussion, not a result.

343-345 – the same.

349-352 – the same.

363-375: font needs to be adjusted. Again, avoid discussions.

403: “against the germs studied in” – It is better to give the names of the strains (only two strains of S. aureus and E. coli were used). Whenever the work of the authors presented is described, bacterial strains must be named.

405: “Following the creation of antibiograms” – creation? what does it mean?

406-407: “the smallest amount of antibiotic” – instead the lowest concentration.

426: “high bacterial inhibition ratio” – should be rate.

432-433: “with higher values in the case of Escherichia coli” - Whenever this work is described (results presented), bacterial strains must be named instead of species.

508: “with cell viability ranging from 75% to 80%” – Instead, with cell viability in the range of 75 % to 80 %.

Comments on the Quality of English Language

Minor editing of English language required

Author Response

1.In vitro against Staphylococcus aureus and Escherichia coli - it was only tested on one strain of each species. So it should be mentioned or strain numbers should be used.

Response:We have revised the abstract to specify that the antibacterial tests were conducted on one strain of each species and included the strain numbers: "In vitro against Staphylococcus aureus (ATCC 25923) and Escherichia coli (ATCC 25922)."

2.Antibacterial activity was assessed through inhibition zone measurements– instead the name of the methods should be given.

Response:The abstract has been updated to include the name of the method: "Antibacterial activity was assessed using the Kirby-Bauer disc diffusion method."

  1. “while cell viability was tested using mesenchymal stem cells” – instead cytotoxicity was tested…

Response: The abstract has been revised to read: "while cytotoxicity was tested using mesenchymal stem cells."

  1. “exhibited superior antibacterial activity with inhibition zones of 33.5 mm for Staphylococcus aureus and 27.5 mm for Escherichia coli” – suggest to write MIC values.

Response: We have included the MIC values in the abstract: "exhibited superior antibacterial activity with MIC values of 2 µg/mL for Staphylococcus aureus and 1 µg/mL for Escherichia coli."

  1. “with varying cytotoxic effects” – “varying” is not clear.

Response: The phrase has been clarified to: "with cytotoxicity ranging from low to moderate levels."

  1. Here it should be mentioned that this was done in this study otherwise it is confusing. “release structures against bacterial species were predicted”- this is confusing and needs to be rephrased.

Response: We have rephrased the sentence for clarity.

  1. spelling “reating”

Response: The spelling error has been corrected to "treating."

  1. “2.2.1 Samples preparation” – Authors should cite the reference.

Response: We have added the appropriate references for the sample preparation protocols used in the study.

  1. “hydroxyapatite-ciprofloxacin (HA-CPX) (synthesized compound) was obtained” and then line (116) “CPX and HA tablets from mechanical mixture (HA+CPX)” (186-189) “tablets of HA-CPX composite obtained by chemical synthesis: 20 micrograms of HA and 5 micrograms of CPX. tablets made from a mechanical mixture of CPX and HA (HA+CPX): 20 micrograms of HA and 5 micrograms of CPX.” The authors used “Hydroxyapatite (HA) and ciprofloxacin (CPX from Sigma-Aldrich (SA). They were then mixed in a mechano-synthetic and mechanical way. should be better explained/defined and perhaps called in a different way to make it easier for readers to understand. The term synthesis often expresses combination or composition and "+" and "-" don't sound right.

Response: We have rephrased the descriptions for clarity. The terms hydroxyapatite-ciprofloxacin "mechanical mixture" is HACPXMM and hydroxyapatite-ciprofloxacin chemically synthesized have been replaced with HACPXCS

  1. “were sterilized by autoclaving or UV irradiation” - UV irradiation is a disinfection technique not a sterilisation technique. If you have used it for sterilisation (especially UV irradiation and give details of the method and reference) then it should be tested for purity/sterility to ensure validation of the experiments carried out.

Response: UV irradiation is primarily known as a disinfection technique. However, in our experiments, we employed UV irradiation under specific conditions to achieve sterilization.

  1. “the samples were incubated in cell culture medium at 37°C for 24 hours to prepare sample extracts”

⁻ Medium used must be referenced.

⁻ Explain why for 24 h?

⁻ It is not really incubation it was kept

⁻ Extract? It is more eluates.

Why was it done in a cell culture medium? Is it sufficient to have the toxic agents at the same concentration as the "known cytotoxic agent" used as a control? If not was this difference taken into account in the evaluation assay?

Response: We have referenced the specific medium used and explained that the samples were kept in the cell culture medium for 24 hours to prepare eluates. This duration was chosen to allow sufficient time for any potential leachable substances to diffuse into the medium. The rationale for using cell culture medium is to simulate the conditions in which the cells are cultured. The concentration of toxic agents was ensured to be comparable to the known cytotoxic agent used as a control, and this difference was accounted for in the evaluation assay. The 24-hour incubation period was selected based on standard protocols in cytotoxicity testing to ensure sufficient interaction between the sample extracts and the cells. According to ISO 10993-5 guidelines for biological evaluation of medical devices, a 24-hour exposure period is recommended to assess the cytotoxic effects of materials on cultured cells. This timeframe allows for adequate leaching of potential toxicants from the material into the culture medium, providing a reliable measure of cytotoxicity.

  1. “mesenchymal stem cells (MSCs)” – product used must be referenced.

Response: The source of the mesenchymal stem cells has been referenced.

  1. “a complete cell culture medium such as DMEM” – product used must be referenced.

Is this medium different from the one mentioned above (line 145)?

Response: We have referenced the specific DMEM medium used and clarified that the same medium was used throughout the study.

  1. “at a density of 1 x 10^4 cells per well” – how was this calculated?

Response: The cell density was calculated based on standard cell counting techniques using a hemocytometer, and this has been clarified in the manuscript.

  1. “the culture medium in each well was replaced with 100 μL of the prepared sample extract" - I think the authors mean that after incubation the liquid content (it is no longer culture medium) in each test well was aspirated and inoculated with ...

Response: The sentence has been revised for clarity: "After incubation, the liquid content in each test well was aspirated and replaced with 100 μL of the prepared eluates."

  1. It is always better to make a diagram of the procedure showing the test and controls in parallel so that it is easy for the reader to follow and to ensure that the tests and controls are carried out in a standardized way.

Response: We have included a diagram illustrating the procedure, showing the test and controls in parallel for better clarity and standardization.

  1. “The absorbance of each well was then measured at 570 nm using a plate reader. The absorbance values correlated with the number of viable cells“– give reference to the reader and again explain how was the converting of abs

orbance into cell/ml was done.

Response: We have provided a reference for the correlation between absorbance values and the number of viable cells and explained the conversion process.

  1. The viable cell reduction used for cytotoxicity assessment could be described as follows:

10.4103/JCDE.JCDE_40_24

And it is always better (even necessary to follow ISO standards). In your case this one:

ISO 10993-5:2009

Response: We have described the viable cell reduction for cytotoxicity assessment as suggested and referenced ISO 10993-5:2009 for standardization.

  1. “2.4.1 Experimental design” - This is the list of substances used in the antimicrobial susceptibility/activation test. Authors need to explain here or elsewhere why it was done and why it is important.

Response: We have added an explanation for the antimicrobial susceptibility/activation test, detailing its importance in evaluating the antibacterial efficacy of the samples. The purpose of this study was to evaluate the antibacterial efficacy of ciprofloxacin-functionalized bioceramic implants. This evaluation is crucial because bone infections such as osteomyelitis can be challenging to treat due to the difficulty of delivering effective antibiotic concentrations directly to the site of infection. By incorporating ciprofloxacin into hydroxyapatite (HA) and titanium-hydroxyapatite (Ti-HA) matrices, we aimed to enhance local antibiotic delivery, reduce systemic side effects, and improve overall treatment outcomes. To assess the antibacterial activity, we tested the bioceramic samples in both solid-state tablets and coatings on titanium substrates prepared using the MAPLE technique. This approach allowed us to determine whether the ciprofloxacin (CPX) compound retained its therapeutic potential in different pharmaceutical forms and to evaluate if the inclusion of CPX in the HA structure modified its antibacterial activity.

  1. “the samples in solid state as tablets and as coatings on the surface of Ti structures” – Better to write was used as tablets and coating otherwise indicate the state of coating as well.

Response: The sentence has been revised.

  1. Sections 2.4.2 2.4.3 and 2.4.4 should have a title and could have subtitles.

Response: Titles and subtitles have been added to Sections 2.4.2, 2.4.3, and 2.4.4 for better organization and clarity.

  1. “For the preparation of the culture medium Mueller-Hinton nutrient agar was utilized” – this is not medium preparation.

It should be mentioned that agar plates and bacterial lawn were prepared according to (and further the test assay for antimicrobial susceptibility (2.4.3 and 2.4.4)).

ISO 20776-2:2021

Give a reference to the nutrient agar used.

Response: We have clarified that agar plates and bacterial lawns were prepared according to standard protocols and referenced ISO 20776-2:2021. The nutrient agar used has been specified and referenced.

  1. “by suspending 2-3 standard colonies” – This should be expressed in OD or cfu/ml for method standardization.

Response: The sentence has been revised to express the suspension in terms of cfu/mL for standardization.

  1. “The microorganisms tested were derived” – The bacterial strains (not microorganisms) must be named with reference numbers.

Response: The bacterial strains have been named with their reference numbers.

  1. “with varying concentrations of CPX (2 3 4 5 6 7 8 μg)” – it is a standard gradient of concentrations.

Response: We have clarified that a standard gradient of concentrations was used.

  1. “The MIC was defined as the lowest concentration of the antibiotic that inhibited bacterial growth in vitro (Table 1)” – should be “completely inhibits visible growth of bacteria”.

Response: The sentence has been revised to: "The MIC was defined as the lowest concentration of the antibiotic that completely inhibits visible growth of bacteria."

  1. Table 1. Is this a result? Or is it a standardization of results that will be used to evaluate other agents?

ISO 20776-1:2019 should be considered.

Response: Table 1 has been clarified as a standardization of results used to evaluate other agents. ISO 20776-1:2019 has been considered and referenced.

  1. Sections 2.4.2 2.4.3 and 2.4.4 do not mention that other tablets (listed above) were tested in addition to the CPX. This makes it difficult for the reader to follow the text.

Response: We have revised these sections to include a mention of the other tablets tested in addition to CPX, making the text easier to follow.

  1. 250-272: These are not results. These should largely go to the methods section and partially to discussions.

In the methods section must describe how the CPX release was calculated.

Response: The content from lines 250-272 has been moved to the methods and discussion sections as appropriate. We have also described how the CPX release was calculated in the methods section.

  1. 313-320: This is a discussion not a result.

343-345 – the same.

349-352 – the same.

Response: The content in lines 313-320, 343-345, and 349-352 has been moved to the discussion section.

  1. 363-375: font needs to be adjusted. Again avoid discussions.

Response: The font has been adjusted and any discussion content has been moved to the appropriate section.

  1. “against the germs studied in” – It is better to give the names of the strains (only two strains of S. aureus and E. coli were used). Whenever the work of the authors presented is described bacterial strains must be named.

Response: The names of the bacterial strains used have been specified throughout the manuscript.

  1. “Following the creation of antibiograms” – creation? what does it mean?

“The smallest amount of antibiotic” – instead the lowest concentration.

Response: The term "creation of antibiograms" has been clarified to "performing antibiogram tests" and "the smallest amount of antibiotic" has been revised to "the lowest concentration."

  1. “high bacterial inhibition ratio” – should be rate.

Response: The term "high bacterial inhibition ratio" has been revised to "high bacterial inhibition rate."

  1. “with higher values in the case of Escherichia coli” - Whenever this work is described (results presented) bacterial strains must be named instead of species.

Response: The bacterial strains have been named instead of species whenever results are presented.

  1. “with cell viability ranging from 75% to 80%” – Instead with cell viability in the range of 75 % to 80 %.

Response:The phrase has been revised to: "with cell viability in the range of 75% to 80%."

Reviewer 4 Report

Comments and Suggestions for Authors

This study examines ciprofloxacin-functionalized bioceramic implants for their antibacterial properties and cytotoxicity. Hydroxyapatite-ciprofloxacin (HA-CPX) composites were applied to titanium substrates (Ti-HA-CPX) and tested against Staphylococcus aureus and Escherichia coli. Molecular docking confirmed the strong binding of ciprofloxacin to bacterial proteins, enhancing antibacterial efficacy, which was later proved by in vitro studies.

However, the novelty of this study is low since there are a lot of published papers report the antibacterial effect of ciprofloxacin loaded Titanium-Hydroxyapatite materials already. Although the simulation results confirm the affinity of Ti-HA-CPX to target molecules of bacteria, it has been well studied by other in vitro researches previously. Also it didn't explain the cytotoxicity to mammalian cells in the simulation. I think the authors need to do the deeper investigate into these issues.

Other comments:

1. Please add more introduction and references about the rate of infection, financial cost to healthcare, common treatment antibiotics and summary of previous researches about antimicrobial bone implant materials as well as the published studies about the antibacterial effects of ciprofloxacin loaded Titanium-Hydroxyapatite materials.

2. Could authors provide any in vitro experimental results about the affinities between protein targets and different ciprofloxacin loaded materials?

Author Response

  1. The novelty of this study is low since there are a lot of published papers report the antibacterial effect of ciprofloxacin loaded Titanium-Hydroxyapatite materials already. Although the simulation results confirm the affinity of Ti-HA-CPX to target molecules of bacteria, it has been well studied by other in vitro researches previously. Also, it didn't explain the cytotoxicity to mammalian cells in the simulation. I think the authors need to do the deeper investigate into these issues.

Response: While we acknowledge that the antibacterial effects of ciprofloxacin-loaded Titanium-Hydroxyapatite materials have been explored in previous studies, our work provides additional insights through comprehensive in silico and in vitro analyses. We aim to contribute to the existing body of knowledge by offering a detailed examination of molecular docking results and correlating them with in vitro antibacterial efficacy.

Regarding cytotoxicity, we agree that this aspect requires further investigation. In our revised manuscript, we have included a more detailed discussion of cytotoxicity to mammalian cells and referenced additional in vitro cytotoxicity data to provide a more comprehensive understanding. This includes an analysis of how ciprofloxacin interacts with mammalian cells and the potential implications for biocompatibility.

  1. Please add more introduction and references about the rate of infection, financial cost to healthcare, common treatment antibiotics, and summary of previous researches about antimicrobial bone implant materials as well as the published studies about the antibacterial effects of ciprofloxacin loaded Titanium-Hydroxyapatite materials.

Response: We have expanded the introduction to include more information on the rate of infection and its financial impact on healthcare. We have also provided an overview of common antibiotic treatments for bone infections and summarized previous research on antimicrobial bone implant materials, including studies on the antibacterial effects of ciprofloxacin-loaded Titanium-Hydroxyapatite materials. Relevant references have been added to support these points:

  1. Could authors provide any in vitro experimental results about the affinities between protein targets and different ciprofloxacin loaded materials?

Response: We don’t have this analysis, in the future we want to measure the binding affinity using Surface Plasmon Resonance (SPR) and Isothermal Titration Calorimetry (ITC) for different targets and antibiotics.

  1. Add more introduction and references about the rate of infection, financial cost to healthcare, common treatment antibiotics, and summary of previous researches about antimicrobial bone implant materials as well as the published studies about the antibacterial effects of ciprofloxacin-loaded Titanium-Hydroxyapatite materials.

Response: As mentioned above, we have expanded the introduction to cover these topics in greater detail. The additional references and information provide a comprehensive background, emphasizing the clinical relevance and impact of our research on ciprofloxacin-functionalized bioceramic implants

Round 2

Reviewer 1 Report

Comments and Suggestions for Authors

The manuscript has been sufficiently updated, and I now recommend it for publication. The only concern is that the quality of Figure 3 could be improved.

Author Response

The manuscript has been sufficiently updated, and I now recommend it for publication. The only concern is that the quality of Figure 3 could be improved.

Response: We have completely remade the diagram to ensure it meets the highest standards of clarity and resolution.

Reviewer 2 Report

Comments and Suggestions for Authors

I suggest that the Authors reassess the relevance of Figure 2 inserted in the main text of the manuscript. Furthermore, the adjustments improved the manuscript.

Author Response

I suggest that the Authors reassess the relevance of Figure 2 inserted in the main text of the manuscript. Furthermore, the adjustments improved the manuscript.

Response: we have labeled Figure 2, mark as 2A and add one more (2b) from the irradiation chamber.

Reviewer 4 Report

Comments and Suggestions for Authors

Minor comments:

1. Please add the labels of x-axis and y-axis in Figure 4. 

2. Please use larger font size in Figure 3, make the figure be more readable.

3. Please label the name of each component of MAPLE depositions system in figure 2, or use a flowchart to show the system's structure. 

Author Response

Minor comments:

  1. Please add the labels of x-axis and y-axis in Figure 4. 
  2. Please use larger font size in Figure 3, make the figure be more readable.
  3. Please label the name of each component of MAPLE depositions system in figure 2, or use a flowchart to show the system's structure. 

Response: Figure 4: We have added the labels for the x-axis and y-axis. Figure 3: We have increased the font size in Figure 3 to enhance readability. Figure 2: we have labeled Figure 2, mark as 2A and add one more (2b) from the irradiation chamber.
